# Investigation of the Amide Linkages on Cooperative Supramolecular Polymerization of Organoplatinum(II) Complexes

**DOI:** 10.3390/molecules26092832

**Published:** 2021-05-10

**Authors:** Mingliang Gui, Yifei Han, Hua Zhong, Rui Liao, Feng Wang

**Affiliations:** CAS Key Laboratory of Soft Matter Chemistry, Department of Polymer Science and Engineering, University of Science and Technology of China, Hefei 230026, China; gml0809@mail.ustc.edu.cn (M.G.); 15200828166@163.com (Y.H.); zh199901@mail.ustc.edu.cn (H.Z.)

**Keywords:** supramolecular polymers, cooperativity, chirality, hydrogen bonds, organoplatinum(II) complexes, self-assembly

## Abstract

Cooperative supramolecular polymerization of π-conjugated compounds into one-dimensional nanostructures has received tremendous attentions in recent years. It is commonly achieved by incorporating amide linkages into the monomeric structures, which provide hydrogen bonds for intermolecular non-covalent complexation. Herein, the effect of amide linkages is elaborately studied, by comparing supramolecular polymerization behaviors of two structurally similar monomers with the same platinum(II) acetylide cores. As compared to the *N*-phenyl benzamide linkages, *N*-[(1*S*)-1-phenylethyl] benzamide linkages give rise to effective chirality transfer behaviors due to the closer distances between the chiral units and the platinum(II) acetylide core. They also provide stronger intermolecular hydrogen bonding strength, which consequently brings higher thermo-stability and enhanced gelation capability for the resulting supramolecular polymers. Supramolecular polymerization is further strengthened by varying the monomers from monotopic to ditopic structures. Hence, with the judicious modulation of structural parameters, the current study opens up new avenues for the rational design of supramolecular polymeric systems.

## 1. Introduction

Supramolecular polymerization of π-conjugated molecules has become an important way to access dynamic functional materials [1,2,3,4,5,6]. To achieve control over the size and shape of the resulting nanostructures, it is important to elucidate the supramolecular polymerization mechanism. Depending on the energy evolution of the supramolecular polymerization processes, three different types of mechanisms, namely isodesmic, cooperative, and anti-cooperative mechanisms, have been clarified [7,8]. The cooperative supramolecular polymerization mechanism, undergoing an initial nucleation step followed by the energetically favorable elongation stage, prevents the formation of small-sized oligomers commonly encountered in the isodesmic and anti-cooperative mechanisms. Moreover, by avoiding the spontaneous nucleation process, living supramolecular polymerization can be achieved on the basis of such cooperative assembled systems, thus providing new avenues toward highly-ordered supramolecular homopolymers and block copolymers [9,10,11,12]. Meriting from all these advantages, great efforts have been devoted to develop π-conjugated supramolecular polymeric systems with the cooperative mechanism. A plausible approach to attain this objective is to embed amide or urea units on the outside of π-conjugated cores. It facilitates the incorporation of intermolecular hydrogen bonds between the neighboring monomers, and thereby induces electronic cooperativity during the supramolecular polymerization process [13,14,15,16,17]. By adopting this principle, a variety of π-conjugated moieties such as arenes, oligo(phenylene ethynylene)s (OPEs), perylene bisimides (PBIs), porphyrins have been reported to form cooperative supramolecular polymers [18,19,20,21,22,23,24,25,26,27,28,29,30].

Organoplatinum(II) complexes are regarded as an intriguing type of π-conjugated compounds [31,32,33,34]. The presence of heavy metal not only increases π-conjugation length, but strengthens spin–orbit coupling with the neighboring ligands, thus endowing organoplatinum(II) complexes with fascinating photoelectrical properties [35,36,37]. More importantly, thanks to the square planar geometry of the *d*^8^ transition metal ion, organoplatinum(II) structures are prone to aggregate with each other by means of π-π stacking and/or metal-metal interactions [38,39]. Our research group has incorporated amides on both ends of platinum(II) acetylide rods (a typical structure is (*S*)-**1** shown in Scheme 1) [40], which demonstrate the adoption of two-stage nucleation-elongation mechanism for the supramolecular polymerization process. By taking advantage of high molecular weight, ease of processability and strong emission properties for these supramolecular polymers, we have further demonstrated their applications in optical waveguide and anti-counterfeiting materials [41,42,43].

In the previous organoplatinum(II)-based supramolecular systems [41,42,43], intermolecular hydrogen bonds are formed on the basis of *N*-phenylbenzamide linkages. Herein, a novel platinum(II) acetylide monomer (*S*)-**2** has been designed (Scheme 1) and synthesized (Scheme 2), which contains two *N*-[(1*S*)-1-phenylethyl]benzamide linkages. We envisage that the structural variations might influence intermolecular hydrogen bonding strength. As a consequence, distinct supramolecular polymerization behaviors occur between the two structurally similar monomers (*S*)-**1** and (*S*)-**2**. Another interesting point for (*S*)-**2** is the simultaneous incorporation of chiral methyl fragments into the amide linkages, which facilitates to the formation of single-handed chiral supramolecular polymers. As compared to those of (*S*)-**1**, the chiral units of (*S*)-**2** move closer to the platinum(II) acetylide core, and thereby result in stronger chirality transfer capabilities. Accordingly, the elaborate manipulation of structural parameters at molecular level opens up new avenues for the rational construction of supramolecular polymeric systems.

## 2. Results

### 2.1. Supramolecular Polymerization Behaviors of (S)-**2** in Apolar Media

As a first step, supramolecular assembly of monomer (*S*)-**2** in apolar methylcyclohexane (c = 4.00 × 10*^−^*^5^ mol/L, MCH) was studied through UV–Vis and circular dichroism (CD) spectroscopy. At 323 K, it exhibited a high-energy absorbance band between 250 and 307 nm (λ_max_: 265 nm, ε = 5.10 × 10^4^ L cm*^−^*^1^ mol*^−^*^1^), together with a low-energy absorption band between 310 and 368 nm (λ_max_: 335 nm, ε = 3.18 × 10^4^ L cm*^−^*^1^ mol*^−^*^1^, Figure 1a). According to the previous literature, the former band corresponds to π-π* transition of the chiral N-[(1S)-1-phenylethyl] benzamide moieties [44]. Meanwhile, the latter band is assigned to PEt_3_-based intra-ligand (IL) π-π* transition, together with some admixture of metal d orbitals [45,46]. In the CD spectrum, a very weak Cotton effect below 330 nm was observed at 323 K, which derives from the intrinsic molecular chirality (Figure 1a). Accordingly, at high temperature (*S*)-**2** is dominated by a molecularly dissolved state in apolar media.

Upon lowering the temperature to 283 K, a bathochromic shift was observed for the IL transition band (λ_max_ = 342 nm, Figure 1a). Moreover, two isosbestic points exist at 298 and 335 nm, suggesting the reversible transition between the monomeric and aggregated states of (*S*)-**2** upon varying the temperature (Appendix A). In the meantime, obvious Cotton effects appeared for (*S*)-**2** at the low-energy IL absorption region, with the positive maximum at 349 nm (Δε = 157 L cm^−1^ mol^−1^, g = 0.00623) (Figure 1a). Mirror imaged CD signals were observed for the enantiomer (*R*)-**2** (Figure 1a). As widely documented, CD spectroscopy is a powerful technique to probe the regularity of the self-assembled chiral structures, since the induced CD signals are only present for long-range ordered assemblies yet absent for short-ranged oligomeric counterparts [47,48]. Accordingly, these experimental results validate chirality transfer from the *N*-[(1*S*)-1-phenylethyl]benzamide linkages to the inner platinum(II) acetylide core, leading to the formation of single-handed supramolecular polymers at low temperature. It is worthy to note that the measured CD signals were real to reflect the supramolecular chirality, since no linear dichroism (LD) signals were detected under the same circumstances (Appendix A) [49]. The formation of long-range ordered supramolecular polymeric structures for (*S*)-**2** was further demonstrated by transmission electron microscope (TEM) experiments, which showed the intertwined nanofibers with several microns in length and approximately 70 nm in width (Figure 1b and Appendix A).

Interestingly, supramolecular chirality signals of (*S*)-**2** differ from those of (*S*)-**1**. In particular, unlike the presence of postive CD signals for (*S*)-**2**, (*S*)-**1** exhibited bisignate CD signals in the IL absorption region at 283 K, with the positive maximum at 349 nm and the negative one at 323 nm (Figure 1c). Moreover, the Cotton effects of (*S*)-**1** were much lower than those of (*S*)-**2**, even cooling the MCH solution of (*S*)-**1** to 273 K (at which point the monomers reach fully aggregated, vide infra) (Δε: –55.2 L cm*^−^*^1^ mol*^−^*^1^ for (*S*)-**1** at 374 nm versus 157 L cm*^−^*^1^ mol*^−^*^1^ for (*S*)-**2** at 349 nm, Figure 1c). It is rationalized that (*S*)-**2** features closer distances between the chiral (1*S*)-1-phenylethyl units and the platinum(II) acetylide rod. As a consequence, it gives rise to stronger capability for supramolecular chiralilty transfer. The results were similar to the previous BTA-based supramolecular systems reported by Meijer and coworkers, which exhibited strong CD signals when the peripherial chiral units move closer to the π-aromatic core [50].

### 2.2. Comparison of Supramolecular Polymerization Thermodynamics between (S)-**1** and (S)-**2**

Deeper insights into the supramolecular polymerization behaviors of (*S*)-**2** were achieved via temperature-dependent UV–Vis and CD experiments. Non-sigmoidal melting curves were obtained, by monitoring the absorption intensities at 348 nm (Figure 1d) and CD intensities at 350 nm (Appendix A) with the melting rate at 60 K h^−1^. The phenomena supported the adoption of cooperative nucleation-elongation mechanism for the supramolecular polymerization process. The normalized melting curves were further fitted by the Meijer-Schenning-Van der Schoot mathematical model (see Equation (1) in the Section 4) [51,52]. Depending on the UV–Vis melting curve, *T*_e_ (critical elongation temperature) value of (*S*)-**2** was determined to be 299.7 K at the monomer concentration of 4.00 × 10^−5^ mol/L in MCH solution, while the *h*_e_ (enthalpy release upon elongation) value was –96.5 kJ mol^−1^ (Figure 1d and Appendix A). Similar values were obtained for the CD melting curve under the same circumstances (*T*_e_: 298.7 K, *h*_e_: –93.5 kJ mol^−1^, Appendix A). On this basis, the quantitative supramolecular polymerization parameters were acquired via the modified Van’t Hoff plots (Figure 1e). As shown in Table 1, the enthalpy release (Δ*H*) value and the entropy release (Δ*S*) value were determined to be –95.1 kJ mol^−1^ and –233 J mol^−1^ K^−1^ in MCH, giving rise to the Gibbs free energy (Δ*G*) value of –26.8 kJ mol^−1^ at 293 K (Table 1, Appendix A).

We have previously reported that supramolecular polymerization of (*S*)-**1** also adopts nucleation-elongation cooperation mechanism. Nevertheless, the thermodynamic parameters of the two compounds are different. In particular, (*S*)-**2** displays more negative Δ*H* and Δ*G* values than those of (*S*)-**1** (Δ*H:* –95.1 kJ mol^−1^ vs. –81.0 kJ mol^−1^; Δ*G:* –26.8 kJ mol^−1^ vs. –22.9 kJ mol^−1^ at 293 K, Table 1, Appendix A). Accordingly, it indicates stronger non-covalent complexation strength for the former compound during the supramolecular polymerization process. Such properties consequently bring higher thermo-stability for supramolecular polymers, as evidenced by the increased *T*_e_ values [299.7 K for supramolecular polymers derived from (*S*)-**2** vs. 287.0 K for those from (*S*)-**1** at the concentration of 0.04 mM in MCH, Figure 1d]. Furthermore, (*S*)-**2** displayed strong gelation capability in MCH. The critical gelation concentration (CGC) of (*S*)-**2** was 10.9 mM at room temperature (Figure 1b, inset). According to the rheological experiments, the storage modulus *G*’ and loss modulus *G*’’ of (*S*)-**2** gels in MCH were determined to be 4755 Pa and 713 Pa, respectively (Appendix A). In sharp contrast, (*S*)-**1** failed to gelate MCH at room temperature, and formed organogels in heptane with a higher CGC value (37.0 mmol L^–1^, Appendix A) [40].

The distinct properties between (*S*)-**1** and (*S*)-**2** emphasize the amide linkage effects on supramolecular polymerization behaviors. The reasonable explanation was achieved by density functional theory (DFT) computations. The NH---O bond lengths between the neighboring amide units were determined to be 1.88 Å and 1.89 Å in (*S*)-**2**_2_, which were shorter than those of (*S*)-**1**_2_ (2.03 Å and 2.02 Å, Figure 2). Besides, the Gibbs free energy (Δ*G*) value of (*S*)-**2**_2_ was more negative than that of (*S*)-**1**_2_ (−159 kJ mol^−1^ vs. −138 kJ mol^−1^, Figure 2). Hence, intermolecular hydrogen-bond interactions of (*S*)-**2** were stronger than those of (*S*)-**1**, which exert crucial impacts on the stability of supramolecular polymers. The participation of intermolecular hydrogen bonds in (*S*)-**2** was further verified by means of ^1^H NMR measurements (Appendix A), which showed downfield shift of NH protons upon increasing the monomer concentration (from 6.13 ppm at 2.00 mM to 6.19 ppm at 30.0 mM in deuterio-chloroform).

### 2.3. Majority-Rules for Supramolecular Polymers Derived from (S)-**2**/(R)-**2**

In supramolecular polymeric systems, chirality amplification is increasingly important for both fundamental science and practical applications [53,54,55,56]. It denotes the transfer of chiral information from molecular to supramolecular levels, in which “majority-rules” effects refer to the ability of a small enantiomeric imbalance to dictate the handedness of helical polymers [57,58,59]. With both (*S*)-**2** and (*R*)-**2** in hand, majority-rules experiments were performed by mixing the two enantiomers in different ratios at the total concentration of 4.00 × 10^−5^ mol/L at 293 K. The full CD spectra showed the transition from pure (*R*)-**2** (*ee* = 100%) through the (*S*)-**2**/(*R*)-**2** (1:1) (*ee* = 0%) to pure (*S*)-**2** (*ee* = –100%) (Figure 3a). By probing the net helicity at 351 nm *versus* the *ee* values, an almost linear plot was obtained (Figure 3b), illustrating rather weak majority-rules effect for (*S*)-**2**/(*R*)-**2**. Accordingly, it reflected the strong homo-recognition between monomers (*S*)-**2** and (*R*)-**2**, thus giving rise to narcissistic self-sorted supramolecular homopolymers. Additional evidence for the absence of hetero-complexation between (*S*)-**2** and (*R*)-**2** was deduced from the melting curves of (*S*)-**2**/(*R*)-**2** (*ee* = −50%) at a total concentration of 4.00 × 10^−5^ mol/L. In particular, the elongation temperature *T*_e_ was determined to be 296.4 K (Appendix A). The value is lower than that of pure (*S*)-**2** at 4.00 × 10^−5^ mol/L (*T*_e_ = 298.7 K, Appendix A), while similar to that of (*S*)-**2** at 3.00 × 10^−5^ mol/L (*T*_e_ = 297.6 K, Appendix A). Therefore, it verified poor mixing capability for the opposite enantiomers, thus giving rise to self-sorted supramolecular homopolymers with low thermal stability.

The absence of majority-rules effects has been previously reported in some supramolecular polymeric systems [60,61,62]. For example, Meijer’s group developed zinc porphyrin-based supramolecular polymers with (*S* or *R*)-3,7-dimethyloctyl side chains on the monomeric structures. Ascribed to the presence of twelve methyl stereogenic centers, high structural mismatching exists. Consequently, a chiral monomer fails to incorporate into a stack of its opposite chirality, and ultimately gives rise to narcissistic self-sorting into conglomerate stacks. In the current system, structural mismatching also exists between the two enantiomers (*S*)-**2** and (*R*)-**2**, despite the fact that only two methyl stereogenic centers are present in the monomeric structures. We envisaged that the *N*-[(1*S*)-1-phenylethyl] benzamide linkages impose large steric hindrance around the chiral centers, and thereby disfavors hetero-complexation between (*S*)-**2** and (*R*)-**2**. Besides, the presence of large steric Pt(PEt_3_)_2_ unit might also decrease the likelihood of coaggregation between the opposite enantiomers.

### 2.4. Supramolecular Polymerization Behaviors of (S)-**3**

In addition to the monotopic monomer (*S*)-**2**, we also studied supramolecular polymerization behaviors of the ditopic monomer (*S*)-**3**. It is not soluble in pure MCH. The phenomenon is ascribed to the presence of four amide units, which provide stronger aggregation tendency than that of (*S*)-**2**. In this regard, UV–Vis and CD spectroscopic measurements were performed for (*S*)-**3** in MCH/TCE (99:1, *v*/*v*, *c* = 2.00 × 10^−5^ mol/L). For UV–Vis spectrum of (*S*)-**3** at 293 K, it exhibited two absorption bands centered at 264 and 333 nm, respectively (Appendix A), which were similar to those of (*S*)-**2**. Cotton effects were observed for (*S*)-**3** in MCH/TCE (99:1, *v*/*v*, *c* = 2.00 × 10^−5^ mol/L), with the positive maximum at 342 nm (Δ*ε* = 102 L cm^−^^1^ mol^−^^1^, *g* = 0.00314) and negative maximum at 321 nm (Δ*ε* = –16.6 L cm^−^^1^ mol^−^^1^, *g* = –0.000465) (Figure 4b). Depending on temperature-dependent CD measurements, a non-sigmoidal melting curve was obtained by monitoring the CD intensities at 342 nm, supporting the adoption of nucleation-elongation cooperative mechanism (Appendix A). Notably, the *T*_e_ value of (*S*)-**3** was determined to be 302.8 K (Figure 4c), which was significantly higher than the *T*_e_ value of (*S*)-**2** (290.4 K) at the same amounts of platinum acetylide units (MCH/TCE = 99:1, *c* = 4.00 × 10^–5^ mol/L, Figure 4c). It is rationalized that the octadecane unit in (*S*)-**3** serves as the crosslinking unit, leading to the formation of supramolecular polymeric networks (Figure 4a). It consequently leads to stronger gelation capability: the CGC value of (*S*)-**3** was determined to be 2.47 mM in MCH/TCE (98:2) (Figure 4a), which is approximately four times lower than that of (*S*)-**2** (10.9 mM) in pure MCH.

## 3. Discussion

In summary, the amide linkage effects have been investigated, by comparing supramolecular polymerization behaviors of two structurally similar monomers (*S*)-**1** and (*S*)-**2.** Both of them undergo chiral supramolecular polymerization in apolar media via the nucleation-elongation cooperative mechanism. Interestingly, as compared to the *N*-phenyl benzamide linkages in (*S*)-**1**, *N*-[(1*S*)-1-phenylethyl] benzamide linkages in (*S*)-**2** give rise to effective chirality transfer behaviors, thanks to the closer distances between the chiral units and the platinum(II) acetylide rod. They also provide stronger intermolecular hydrogen bonding strength, which consequently brings higher thermo-stability and enhanced gelation capability for the resulting supramolecular polymers. Majority-rules effects are absent between (*S*)-**2** and the enantiomer (*R*)-**2**, primarily ascribed to the large steric hindrance rendered by *N*-[(1*S*)-1-phenylethyl] benzamide linkages around the chiral center. Moreover, supramolecular polymerization and gelation capabilities are further strengthened, by varying the structures from the monotopic monomer (*S*)-**2** to ditopic one (*S*)-**3**. Therefore, the current study blooms new avenues for the rational design of supramolecular polymeric systems with tailored properties.

## 4. Materials and Methods

### 4.1. Materials and Instruments

Pd(PPh_3_)_2_Cl_2_, CuI, potassium carbonate (K_2_CO_3_) and (trimethylsilyl)acetylene were reagent grade and were employed as received. All chemical reagents and solvents are commercially available. ^1^H-NMR spectra were collected on a Ascend^TM^ 400 MHz spectrometer (Bruker, Karlsruhe, Germany) with trimethylsilane as the internal standard. ^13^C-NMR spectra were recorded on a Bruker Ascend^TM^ 400 MHz spectrometer at 101 MHz. Time-of-flight mass spectra (TOF−MS) were obtained on matrix−assisted laser desorption ionization−time of flight (autoflex speed TOF/TOF, Bruker). UV−vis spectra were recorded on a UV-1800 spectrometer (Shimadzu, Kyoto, Japan). CD measurements were performed on a J-1500 CD spectrometer (Jasco, Kyoto, Japan) equipped with a PFD-425S/15 Peltier-type temperature controller. The gel sample was transferred onto the plate at 298 K. Oscillatory dynamic shear experiments were performed in the frequency range of 0.2–200 rad s^−1^, using a constant strain (0.2%) determined with a strain sweep to lie within the linear viscoelastic regime.

### 4.2. DFT Computations

DFT computations were performed on the Gaussian 09, version D.01, software package (Wallingford, CT, USA). During the optimization process, nonmetallic atoms were described by the wb97xd/6-31G(d) computational level, while all platinum(II) atoms were described by the Lanl2dz effective core potential.

### 4.3. Determination of the Assembling Thermodynamics for the Supramolecular Polymerization Process

Compounds (*S*)-**1**, (*S*)-**2** and (*S*)-**3** adopt the cooperative supramolecular polymerization mechanism. Therefore, normalized UV−vis and CD melting curves were fitted with the Meijer−Schenning−van der Schoot mathematical model to obtain thermodynamic parameters in the supramolecular assembly process. In detail, the entire process can be divided into two separate steps: the nucleation and elongation regimes.

In the elongation regime, the fraction of aggregated molecules (*φ*_n_) is described by Equation (1):*φ*_n_ = *φ*_SAT_{1 − exp[(–*h*_e_) × (*T* − *T*_e_)/(*R* × *T*_e_^2^)]}(1)

In this equation, *h*_e_ denotes the molecular enthalpy release due to the non-covalent supramolecular polymerization, *T* and *T*_e_ stand for the absolute temperature and elongation temperature, respectively. *R* represents the universal gas constant. *φ*_SAT_ is a parameter that is introduced to prevent the relation *φ*_n_*/φ*_SAT_ surpassing the value of one.

### 4.4. Synthetic Procedures

#### 4.4.1. Synthesis of (*S*)-**2**

Compound **6** (250 mg, 0.31 mmol), *trans*-PtI_2_(PEt_3_)_2_ (85.0 mg, 0.12 mmol) and CuI (4.70 mg, 0.20 mmol) were mixed in triethylamine/CH_2_Cl_2_ (10 mL:20 mL) under a nitrogen atmosphere and stirred at room temperature for 24 h. The solvents were evaporated on a rotary evaporator. The residue was purified by flash column chromatography (silica gel, CH_3_OH/CH_2_Cl_2_, 1:99 *v/v* as the eluent) to provide (*S*)-**2** as a yellow solid (210 mg, 83%). ^1^H-NMR (400 MHz, CDCl_3_, 298 K): *δ* 7.19 (m, 8H), 6.86 (s, 4H), 6.07 (d, *J* = 7.7 Hz, 2H), 5.20 (m, 2H), 3.91 (m, 12H), 2.09 (m, 12H), 1.70 (m, 12H), 1.51 (d, *J* = 7.8 Hz, 6H), 1.38 (m, 12H), 1.30–1.17 (m, 96H), 1.13 (m, 18H), 0.81 (t, *J* = 6.7 Hz, 18H). ^13^C-NMR (101 MHz, CDCl_3_, 298 K): *δ* 166.42, 153.11, 141.21, 131.78, 131.10, 129.52, 128.04, 126.10, 105.76, 73.52, 69.48, 49.10, 31.96, 30.32, 29.89–29.53, 29.41, 26.10, 22.73, 21.39, 16.48, 14.16, 8.35. MALDI-TOF ([M + H]^+^): *m*/*z* 2033.9165.

#### 4.4.2. Synthesis of (*S*)*-***3**

Compound **7** (270 mg, 0.20 mmol) and CuI (3.42 mg, 0.02 mmol) were mixed in Et_3_N/CH_2_Cl_2_ (20 mL:30 mL) under a nitrogen atmosphere. Compound **8** (71.0 mg, 0.09 mmol) in CH_2_Cl_2_ (10 mL) was added dropwise to the reaction solution. After stirring at room temperature for 24 h, the solvents were evaporated. The residue was purified by flash column chromatography (silica gel, CH_3_OH/CH_2_Cl_2_, 1:99 *v/v* as the eluent) to provide (*S*)-**3** as a yellow solid (210 mg, 72%). ^1^H-NMR (400 MHz, CDCl_3_, 298 K): *δ* 7.70 (d, *J* = 8.7 Hz, 4H), 7.31–7.17 (m, 16H), 6.93 (s, 4H), 6.88 (d, *J* = 8.7 Hz, 4H), 6.17 (dd, *J* = 10.6, 7.7 Hz, 4H), 5.26 (m, 4H), 3.98 (m, 16H), 2.17 (m, 24H), 1.76 (m, 16H), 1.66–1.52 (m, 16H), 1.47–1.23 (m, 132H), 1.19 (t, *J* = 8.0 Hz, 36H), 0.88 (t, *J* = 6.7 Hz, 18H). ^13^C-NMR (101 MHz, CDCl_3_, 298 K): *δ* 175.62, 166.41, 166.08, 161.77, 153.09, 141.18, 139.80, 139.56, 131.74, 131.09, 129.92, 129.56, 128.67, 128.01, 126.57, 126.08, 125.98, 114.20, 105.77, 73.51, 69.44, 68.20, 49.02, 35.96, 31.95, 30.32, 29.93–29.50, 29.39, 29.27, 29.16, 27.23, 26.06, 25.57, 22.72, 21.55, 21.32, 16.33, 14.16, 8.36. MALDI-TOF ([M + H]^+^): *m*/*z* 3240.7299.

## Data Availability

The data presented in this study are available included in the article and Appendix A.

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
