# Peer review of "Investigation of the Amide Linkages on Cooperative Supramolecular Polymerization of Organoplatinum(II) Complexes"

_molecules, 2021, doi:10.3390/molecules26092832_

Round 1

Reviewer 1 Report

In this manuscript, it is described that intermolecular interaction between the chiral acetylide which coordinates to platinum-phosphine moieties leads chiral assembly polymer and thus the solution of the compounds shows temperature dependency on intensity of CD spectra.

The reaction is attractive and obtained products are also characterized definitely. The interpretations of CD spectra are also appropriate, I think.

 I have one disagreeing thing, if it may be minor thing. In line 187 and Figure 3., the author said that linear plot was obtained as function of the ee about the net helicity. However, I can regard this correlation as not linear but rather sigmoid. Is it necessary the correlation is linear?

Author Response

  1. Reply to the comment made by Reviewer 1 “I have one disagreeing thing, if it may be minor thing. In line 187 and Figure 3., the author said that linear plot was obtained of the ee about the net helicity. However, I can regard this correlation as not linear but rather sigmoid. Is it necessary the correlation is linear?

Thank the reviewer for the kind suggestion! The majority-rules effect is weak when the net helicity is linear, while it is strong when the net helicity deviates from linearity. In the current system, the net helicity only shows a very small deviation from linearity, suggesting rather weak majority-rules effect.

Reviewer 2 Report

In this paper, the authors investigated the cooperative supramolecular polymerization of π-conjugated compounds by incorporating amide linkages into the monomeric structures. This work is interesting and well planned with a complex description of synthesis and characterization of the polymeric systems from molecular to supramolecular levels, and I will recommend its publication after minor revisions. There are some suggestions as follows:

  1. Section 4.1 should cover the brief description of MALDI-TOF measurements. In addition, I suggest providing, if it is possible MALDI-TOF measurements for (S)-3.
  2. Is it possible to provide the CD measurements in the magnetic field (MCD) at least for the best case here? (this is not mandatory at present, however, is highly recommended for future study).

Author Response

  1. Reply to the first comment made by Reviewer 2 “Section 4.1 should cover the brief description of MALDI−TOF measurements. In addition, I suggest providing, if it is possible MALDI−TOF measurements for (S)-3.

According to the reviewer’s comment, we have added the description of MALDI−TOF measurements in Section 4.1 [Time-of-flight mass spectra (TOF−MS) were obtained on matrix−assisted laser desorption ionization−time of flight (autoflex speed TOF/TOF, Bruker)]. Meanwhile, we have supplemented the MALDI−TOF measurements for compound (S)-3 in the manuscript.

  1. Reply to the second comment made by Reviewer 2 “Is it possible to provide the CD measurements in the magnetic field (MCD) at least for the best case here? (this is not mandatory at present, however, is highly recommended for future study).”

Thank the reviewer for the kind suggestion! Currently we are unable to perform the MCD measurements due to the unavailability of this apparatus in our university. We will try to contact the potential collaborator in the MCD field in the future study.